# Adaptive Density Estimation for Generative Models

**Thomas Lucas**[*]
Inria[†]

**Konstantin Shmelkov**[*,‡]
Noah's Ark Lab, Huawei

**Cordelia Schmid**
Inria[†]

**Karteek Alahari**
Inria[†]

**Jakob Verbeek**
Inria[†]

## Abstract

Unsupervised learning of generative models has seen tremendous progress over recent years, in particular due to generative adversarial networks (GANs), variational autoencoders, and flow-based models. GANs have dramatically improved sample quality, but suffer from two drawbacks: (i) they mode-drop, *i.e.*, do not cover the full support of the train data, and (ii) they do not allow for likelihood evaluations on held-out data. In contrast, likelihood-based training encourages models to cover the full support of the train data, but yields poorer samples. These mutual shortcomings can in principle be addressed by training generative latent variable models in a hybrid adversarial-likelihood manner. However, we show that commonly made parametric assumptions create a conflict between them, making successful hybrid models non trivial. As a solution, we propose to use deep invertible transformations in the latent variable decoder. This approach allows for likelihood computations in image space, is more efficient than fully invertible models, and can take full advantage of adversarial training. We show that our model significantly improves over existing hybrid models: offering GAN-like samples, IS and FID scores that are competitive with fully adversarial models, and improved likelihood scores.

## 1 Introduction

Successful recent generative models of natural images can be divided into two broad families, which are trained in fundamentally different ways. The first is trained using likelihood-based criteria, which ensure that all training data points are well covered by the model. This category includes variational autoencoders (VAEs) [25, 26, 39, 40], autoregressive models such as PixelCNNs [46, 53], and flow-based models such as Real-NVP [9, 20, 24]. The second category is trained based on a signal that measures to what extent (statistics of) samples from the model can be distinguished from (statistics of) the training data, *i.e.*, based on the quality of samples drawn from the model. This is the case for generative adversarial networks (GANs) [2, 15, 22], and moment matching methods [28].

Despite tremendous recent progress, existing methods exhibit a number of drawbacks. Adversarially trained models such as GANs do not provide a density function, which poses a fundamental problem as it prevents assessment of how well the model fits held out and training data. Moreover, adversarial models typically do not allow to infer the latent variables that underlie observed images. Finally, adversarial models suffer from mode collapse [2], *i.e.*, they do not cover the full support of the training data. Likelihood-based model on the other hand are trained to put probability mass on all elements of the training set, but over-generalise and produce samples of substantially inferior quality as compared to adversarial models. The models with the best likelihood scores on held-out data are autoregressive models [35], which suffer from the additional problem that they are extremely

---

[*] The authors contributed equally.　[†] Univ. Grenoble Alpes, Inria, CNRS, Grenoble INP, LJK, 38000 Grenoble, France.　[‡] Work done while at Inria.

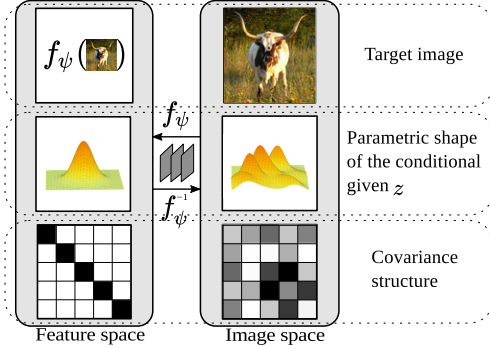

Figure 1: An invertible non-linear mapping $f_\psi$ maps an image $x$ to a vector $f_\psi(x)$ in feature space. $f_\psi$ is trained to adapt to modelling assumptions made by a trained density $p_\theta$ in feature space. This induces full covariance structure and a non-Gaussian density.

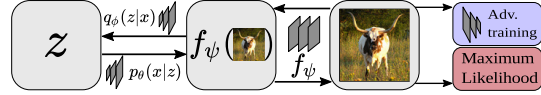

Figure 2: Variational inference is used to train a latent variable generative model in feature space. The invertible mapping $f_\psi$ maps back to image space, where adversarial training can be performed together with MLE.

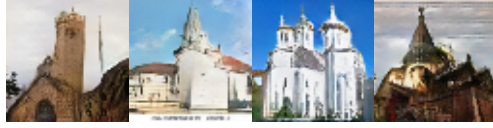

Figure 3: Our model yields compelling samples while the optimization of likelihood ensures coverage of all modes in the training support and thus sample diversity, here on LSUN churches ($64 \times 64$).

inefficient to sample from [38], since images are generated pixel-by-pixel. The sampling inefficiency makes adversarial training of such models prohibitively expensive.

In order to overcome these shortcomings, we seek to design a model that (i) generates high-quality samples typical of adversarial models, (ii) provides a likelihood measure on the entire image space, and (iii) has a latent variable structure to enable efficient sampling and to permit adversarial training. Additionally we show that, (iv) a successful hybrid adversarial-likelihood paradigm requires going beyond simplifying assumptions commonly made with likelihood based latent variable models. These simplifying assumptions on the conditional distribution on data $x$ given latents $z$, $p(x|z)$, include full independence across the dimensions of $x$ and/or simple parametric forms such as Gaussian [25], or use fully invertible networks [9, 24]. These assumptions create a conflict between achieving high sample quality and high likelihood scores on held-out data. Autoregressive models, such as pixelCNNs [46, 53], do not make factorization assumptions, but are extremely inefficient to sample from. As a solution, we propose learning a non-linear invertible function $f_\psi$ between the image space and an abstract feature space, as illustrated in Figure 1. Training a model with full support in this feature space induces a model in the image space that does not make Gaussianity, nor independence assumptions in the conditional density $p(x|z)$. Trained by MLE, $f_\psi$ adapts to modelling assumptions made by $p_\theta$ so we refer to this approach as "adaptive density estimation".

We experimentally validate our approach on the CIFAR-10 dataset with an ablation study. Our model significantly improves over existing hybrid models, producing GAN-like samples, and IS and FID scores that are competitive with fully adversarial models, see Figure 3. At the same time, we obtain likelihoods on held-out data comparable to state-of-the-art likelihood-based methods which requires covering the full support of the dataset. We further confirm these observations with quantitative and qualitative experimental results on the STL-10, ImageNet and LSUN datasets.

## 2   Related work

Mode-collapse in GANs has received considerable attention, and stabilizing the training process as well as improved and bigger architectures have been shown to alleviate this issue [2, 17, 37]. Another line of work focuses on allowing the discriminator to access batch statistics of generated images, as pioneered by [22, 45], and further generalized by [29, 32]. This enables comparison of distributional statistics by the discriminator rather than only individual samples. Other approaches to encourage diversity among GAN samples include the use of maximum mean discrepancy [1], optimal transport [47], determinental point processes [14] and Bayesian formulations of adversarial training [42] that allow model parameters to be sampled. In contrast to our work, these models lack an inference network, and do not define an explicit density over the full data support.

An other line of research has explored inference mechanisms for GANs. The discriminator of BiGAN [10] and ALI [12], given pairs $(x, z)$ of images and latents, predict if $z$ was encoded from a real image, or if $x$ was decoded from a sampled $z$. In [52] the encoder and the discriminator are collapsed into one network that encodes both real images and generated samples, and tries to

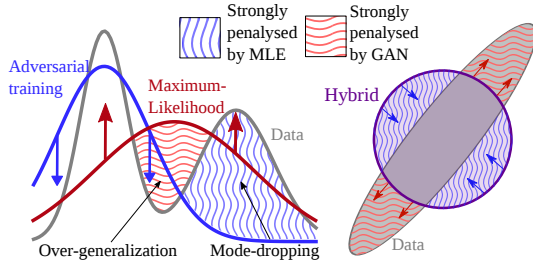

Figure 4: (Left) Maximum likelihood training pulls probability mass towards high-density regions of the data distribution, while adversarial training pushes mass out of low-density regions. (Right) Independence assumptions become a source of conflict in a joint training setting, making hybrid training non-trivial.

spread their posteriors apart. In [6] a symmetrized KL divergence is approximated in an adversarial setup, and uses reconstruction losses to improve the correspondence between reconstructed and target variables for $x$ and $z$. Similarly, in [41] a discriminator is used to replace the KL divergence term in the variational lower bound used to train VAEs with the density ratio trick. In [33] the KL divergence term in a VAE is replaced with a discriminator that compares latent variables from the prior and the posterior in a more flexible manner. This regularization is more flexible than the standard KL divergence. The VAE-GAN model [27] and the model in [21] use the intermediate feature maps of a GAN discriminator and of a classifier respectively, as target space for a VAE. Unlike ours, these methods do not define a likelihood over the image space.

Likelihood-based models typically make modelling assumptions that conflict with adversarial training, these include strong factorization and/or Gaussianity. In our work we avoid these limitations by learning the shape of the conditional density on observed data given latents, $p(x|z)$, beyond fully factorized Gaussian models. As in our work, Flow-GAN [16] also builds on invertible transformations to construct a model that can be trained in a hybrid adversarial-MLE manner, see Figure 2.However, Flow-GAN does not use efficient non-invertible layers we introduce, and instead relies entirely on invertible layers. Other approaches combine autoregressive decoders with latent variable models to go beyond typical parametric assumptions in pixel space [7, 18, 31]. They, however, are not amenable to adversarial training due to the prohibitively slow sequential pixel sampling.

# 3  Preliminaries on MLE and adversarial training

**Maximum-likelihood and over-generalization.**  The de-facto standard approach to train generative models is maximum-likelihood estimation. It maximizes the probability of data sampled from an unknown data distribution $p^*$ under the model $p_\theta$ w.r.t. the model parameters $\theta$. This is equivalent to minimizing the Kullback-Leibler (KL) divergence, $\mathcal{D}_{\mathrm{KL}}(p^*||p_\theta)$, between $p^*$ and $p_\theta$. This yields models that tend to cover all the modes of the data, but put mass in spurious regions of the target space; a phenomenon known as "over-generalization" or "zero-avoiding" [4], and manifested by unrealistic samples in the context of generative image models, see Figure 4.Over-generalization is inherent to the optimization of the KL divergence oriented in this manner. Real images are sampled from $p^*$, and $p_\theta$ is explicitly optimized to cover all of them. The training procedure, however, does not sample from $p_\theta$ to evaluate the quality of such samples, ideally using the inaccessible $p^*(x)$ as a score. Therefore $p_\theta$ may put mass in spurious regions of the space without being heavily penalized. We refer to this kind of training procedure as "coverage-driven training" (CDT). This optimizes a loss of the form $\mathcal{L}_{\mathrm{C}}(p_\theta) = \int_x p^*(x) s_c(x, p_\theta)\ \mathrm{d}x$, where $s_c(x, p_\theta) = \ln p_\theta(x)$ evaluates how well a sample $x$ is covered by the model. Any score $s_c$ that verifies: $\mathcal{L}_{\mathrm{C}}(p_\theta) = 0 \iff p_\theta = p^*$, is equivalent to the log-score, which forms a justification for MLE on which we focus.

Explicitly evaluating sample quality is redundant in the regime of unlimited model capacity and training data. Indeed, putting mass on spurious regions takes it away from the support of $p^*$, and thus reduces the likelihood of the training data. In practice, however, datasets and model capacity are finite, and models *must* put mass outside the finite training set in order to generalize. The maximum likelihood criterion, by construction, only measures *how much* mass goes off the training data, not *where* it goes. In classic MLE, generalization is controlled in two ways: (i) inductive bias, in the form of model architecture, controls *where* the off-dataset mass goes, and (ii) regularization controls to which extent this happens. An adversarial loss, by considering samples of the model $p_\theta$, can provide a second handle to evaluate and control where the off-dataset mass goes. In this sense, and in contrast to model architecture design, an adversarial loss provides a "trainable" form of inductive bias.

**Adversarial models and mode collapse.** Adversarially trained models produce samples of excellent quality. As mentioned, their main drawbacks are their tendency to "mode-drop", and the lack of measure to assess mode-dropping, or their performance in general. The reasons for this are two-fold. First, defining a valid likelihood requires adding volume to the low-dimensional manifold learned by GANs to define a density under which training and test data have non-zero density. Second, computing the density of a data point under the defined probability distribution requires marginalizing out the latent variables, which is not trivial in the absence of an efficient inference mechanism. When a human expert subjectively evaluates the quality of generated images, samples from the model are compared to the expert's implicit approximation of $p^*$. This type of objective may be formalized as $\mathcal{L}_Q(p_\theta) = \int_x p_\theta(x)s_q(x, p^*)\,\mathrm{d}x$, and we refer to it as "quality-driven training" (QDT). To see that GANs [15] use this type of training, recall that the discriminator is trained with the loss $\mathcal{L}_{\text{GAN}} = \int_x p^*(x)\ln D(x) + p_\theta(x)\ln(1 - D(x))\,\mathrm{d}x$. It is easy to show that the optimal discriminator equals $D^*(x) = p^*(x)/(p^*(x) + p_\theta(x))$. Substituting the optimal discriminator, $\mathcal{L}_{\text{GAN}}$ equals the Jensen-Shannon divergence,

$$\mathcal{D}_{\text{JS}}(p^*||p_\theta) = \frac{1}{2}\mathcal{D}_{\text{KL}}(p^*||\frac{1}{2}(p_\theta + p^*)) + \frac{1}{2}\mathcal{D}_{\text{KL}}(p_\theta||\frac{1}{2}(p_\theta + p^*)), \qquad (1)$$

up to additive and multiplicative constants [15]. This loss, approximated by the discriminator, is symmetric and contains two KL divergence terms. Note that $\mathcal{D}_{\text{KL}}(p^*||\frac{1}{2}(p_\theta + p^*))$ is an integral on $p^*$, so *coverage driven*. The term that approximates it in $\mathcal{L}_{\text{GAN}}$, *i.e.*, $\int_x p^*(x)\ln D(x)$, is however independent from the generative model, and disappears when differentiating. Therefore, it cannot be used to perform coverage-driven training, and the generator is trained to minimize $\ln(1 - D(G(z)))$ (or to maximize $\ln D(G(z))$), where $G(z)$ is the deterministic generator that maps latent variables $z$ to the data space. Assuming $D = D^*$, this yields

$$\int_z p(z)\ln(1 - D^*(G(z)))\,\mathrm{d}z = \int_x p_\theta(x)\ln\frac{p_\theta(x)}{p_\theta(x) + p^*(x)}\,\mathrm{d}x = \mathcal{D}_{\text{KL}}(p_\theta||(p_\theta + p^*)/2), \quad (2)$$

which is a quality-driven criterion, favoring sample quality over support coverage.

## 4 Adaptive Density Estimation and hybrid adversarial-likelihood training

We present a hybrid training approach with MLE to cover the full support of the training data, and adversarial training as a trainable inductive bias mechanism to improve sample quality. Using both these criteria provides a richer training signal, but satisfying both criteria is more challenging than each in isolation for a given model complexity. In practice, model flexibility is limited by (i) the number of parameters, layers, and features in the model, and (ii) simplifying modeling assumptions, usually made for tractability. We show that these simplifying assumptions create a conflict between the two criteria, making successfull joint training non trivial. We introduce Adaptive Density Estimation as a solution to reconcile them.

Latent variable generative models, defined as $p_\theta(x) = \int_z p_\theta(x|z)p(z)\,\mathrm{d}z$, typically make simplifying assumptions on $p_\theta(x|z)$, such as full factorization and/or Gaussianity, see *e.g.* [11, 25, 30]. In particular, assuming full factorization of $p_\theta(x|z)$ implies that any correlations not captured by $z$ are treated as independent per-pixel noise. This is a poor model for natural images, unless $z$ captures each and every aspect of the image structure. Crucially, this hypothesis is problematic in the context of hybrid MLE-adversarial training. If $p^*$ is too complex for $p_\theta(x|z)$ to fit it accurately enough, MLE will lead to a high variance in a factored (Gaussian) $p_\theta(x|z)$ as illustrated in Figure 4 (right). This leads to unrealistic blurry samples, easily detected by an adversarial discriminator, which then does not provide a useful training signal. Conversely, adversarial training will try to avoid these poor samples by dropping modes of the training data, and driving the "noise" level to zero. This in turn is heavily penalized by maximum likelihood training, and leads to poor likelihoods on held-out data.

**Adaptive density estimation.** The point of view of regression hints at a possible solution. For instance, with isotropic Gaussian model densities with fixed variance, solving the optimization problem $\theta^* \in \max_\theta \ln(p_\theta(x|z))$ is similar to solving $\min_\theta ||\mu_\theta(z) - x||_2$, i.e., $\ell_2$ regression, where $\mu_\theta(z)$ is the mean of the decoder $p_\theta(x|z)$. The Euclidean distance in RGB space is known to be a poor measure of similarity between images, non-robust to small translations or other basic transformations [34]. One can instead compute the Euclidean distance in a feature space, $||f_\psi(x_1) - f_\psi(x_2)||_2$, where $f_\psi$ is chosen so that the distance is a better measure of similarity. A popular way to

obtain $f_\psi$ is to use a CNN that learns a non-linear image representation, that allows linear assessment of image similarity. This is the idea underlying GAN discriminators, the FID evaluation measure [19], the reconstruction losses of VAE-GAN [27] and classifier based perceptual losses as in [21].

Despite their flexibility, such similarity metrics are in general degenerate in the sense that they may discard information about the data point $x$. For instance, two different images $x$ and $y$ can collapse to the same points in feature space, i.e., $f_\psi(x) = f_\psi(y)$. This limits the use of similarity metrics in the context of generative modeling for two reasons: (i) it does not yield a valid measure of likelihood over inputs, and (ii) points generated in the feature space $f_\psi$ cannot easily be mapped to images. To resolve this issue, we chose $f_\psi$ to be a bijection. Given a model $p_\theta$ trained to model $f_\psi(x)$ in feature space, a density in image space is computed using the change of variable formula, which yields $p_{\theta,\psi}(x) = p_\theta(f_\psi(x)) \left| \det \left( \partial f_\psi(x)/\partial x^\top \right) \right|$. Image samples are obtained by sampling from $p_\theta$ in feature space, and mapping to the image space through $f_\psi^{-1}$. We refer to this construction as Adaptive Denstiy Estimation. If $p_\theta$ provides efficient log-likelihood computations, the change of variable formula can be used to train $f_\psi$ and $p_\theta$ together by maximum-likelihood, and if $p_\theta$ provides fast sampling adversarial training can be performed efficiently.

**MLE with adaptive density estimation.** To train a generative latent variable model $p_\theta(x)$ which permits efficient sampling, we rely on amortized variational inference. We use an inference network $q_\phi(z|x)$ to construct a variational evidence lower-bound (ELBO),

$$\mathcal{L}_{\mathrm{ELBO}}^\psi(x, \theta, \phi) = \mathop{\mathbb{E}}_{q_\phi(z|x)} \left[ \ln(p_\theta(f_\psi(x)|z)) \right] - \mathcal{D}_{\mathrm{KL}}(q_\phi(z|x)||p_\theta(z)) \le \ln p_\theta(f_\psi(x)). \quad (3)$$

Using this lower bound together with the change of variable formula, the mapping to the similarity space $f_\psi$ and the generative model $p_\theta$ can be trained jointly with the loss

$$\mathcal{L}_{\mathrm{C}}(\theta, \phi, \psi) = \mathop{\mathbb{E}}_{x \sim p^*} \left[ -\mathcal{L}_{\mathrm{ELBO}}^\psi(x, \theta, \phi) - \ln \left| \det \frac{\partial f_\psi(x)}{\partial x^\top} \right| \right] \ge - \mathop{\mathbb{E}}_{x \sim p^*} \left[ \ln p_{\theta,\psi}(x) \right]. \quad (4)$$

We use gradient descent to train $f_\psi$ by optimizing $\mathcal{L}_{\mathrm{C}}(\theta, \phi, \psi)$ w.r.t. $\psi$. The $\mathcal{L}_{\mathrm{ELBO}}$ term encourges the mapping $f_\psi$ to maximize the density of points in feature space under the model $p_\theta$, so that $f_\psi$ is trained to match modeling assumptions made in $p_\theta$. Simultaneously, the log-determinant term encourages $f_\psi$ to maximize the volume of data points in feature space. This guarantees that data points cannot be collapsed to a single point in the feature space. We use a factored Gaussian form of the conditional $p_\theta(.|z)$ for tractability, but since $f_\psi$ can arbitrarily reshape the corresponding conditional image space, it still avoids simplifying assumptions in the image space. Therefore, the (invertible) transformation $f_\psi$ avoids the conflict between the MLE and adversarial training mechanisms, and can leverage both.

**Adversarial training with adaptive density estimation.** To sample the generative model, we sample latents from the prior, $z \sim p_\theta(z)$, which are then mapped to feature space through $\mu_\theta(z)$, and to image space through $f_\psi^{-1}$. We train our generator using the modified objective proposed by [50], combining both generator losses considered in [15], *i.e.* $\ln[(1 - D(G(z)))/D(G(z))]$, which yields:

$$\mathcal{L}_{\mathrm{Q}}(p_{\theta,\psi}) = - \mathop{\mathbb{E}}_{p_\theta(z)} \left[ \ln D(f_\psi^{-1}(\mu_\theta(z))) - \ln(1 - D(f_\psi^{-1}(\mu_\theta(z)))) \right]. \quad (5)$$

Assuming the discriminator $D$ is trained to optimality at every step, it is easy to demonstrate that the generator is trained to optimize $\mathcal{D}_{\mathrm{KL}}(p_{\theta,\psi}||p^*)$. The training procedure, written as an algorithm in Appendix H, alternates between (i) bringing $\mathcal{L}_{\mathrm{Q}}(p_{\theta,\psi})$ closer to it's optimal value $\mathcal{L}_{\mathrm{Q}}^*(p_{\theta,\psi}) = \mathcal{D}_{\mathrm{KL}}(p_{\theta,\psi}||p^*)$, and (ii) minimizing $\mathcal{L}_{\mathrm{C}}(p_{\theta,\psi}) + \mathcal{L}_{\mathrm{Q}}(p_{\theta,\psi})$. Assuming that the discriminator is trained to optimality at every step, the generative model is trained to minimize a bound on the symmetric sum of two KL divergences: $\mathcal{L}_{\mathrm{C}}(p_{\theta,\psi}) + \mathcal{L}_{\mathrm{Q}}^*(p_{\theta,\psi}) \ge \mathcal{D}_{\mathrm{KL}}(p^*||p_{\theta,\psi}) + \mathcal{D}_{\mathrm{KL}}(p_{\theta,\psi}||p^*) + \mathcal{H}(p^*)$, where the entropy of the data generating distribution, $\mathcal{H}(p^*)$, is an additive constant independent of the generative model $p_{\theta,\psi}$. In contrast, MLE and GANs optimize one of these divergences each.

## 5 Experimental evaluation

We present our evaluation protocol, followed by an ablation study to assess the importance of the components of our model (Section 5.1). We then show the quantitative and qualitative performance of our model, and compare it to the state of the art on the CIFAR-10 dataset in Section 5.2. We present additional results and comparisons on higher resolution datasets in Section 5.3.

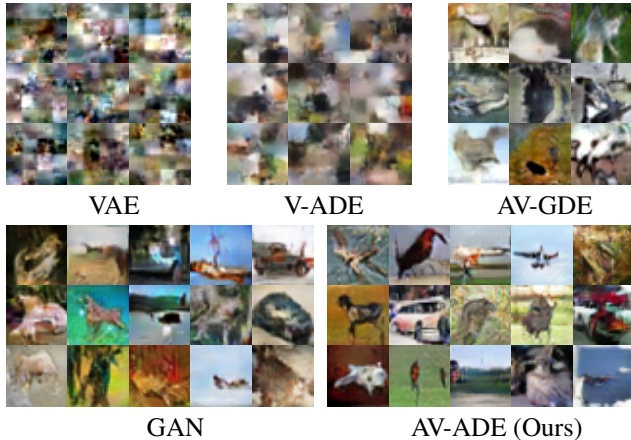

VAE  V-ADE  AV-GDE

GAN  AV-ADE (Ours)

|  | $f_\psi$ | Adv. | MLE | BPD ↓ | IS ↑ | FID ↓ |
|---|---|---|---|---|---|---|
| GAN | × | ✓ | × | [7.0] | 6.8 | 31.4 |
| VAE | × | × | ✓ | 4.4 | 2.0 | 171.0 |
| V-ADE† | ✓ | × | ✓ | 3.5 | 3.0 | 112.0 |
| AV-GDE | × | ✓ | ✓ | 4.4 | 5.1 | 58.6 |
| AV-ADE† | ✓ | ✓ | ✓ | 3.9 | 7.1 | 28.0 |

Table 1: Quantitative results. † : Parameter count decreased by $1.4\%$ to compensate for $f_\psi$. [Square brackets] denote that the value is approximated, see Section 5.

Figure 5: Samples from GAN and VAE baselines, our V-ADE, AV-GDE and AV-ADE models, all trained on CIFAR-10.

**Evaluation metrics.** We evaluate our models with complementary metrics. To assess sample quality, we report the Fréchet inception distance (FID) [19] and the inception score (IS) [45], which are the de facto standard metrics to evaluate GANs [5, 54]. Although these metrics focus on sample quality, they are also sensitive to coverage, see Appendix D for details. To specifically evaluate the coverage of held-out data, we use the standard bits per dimension (BPD) metric, defined as the negative log-likelihood on held-out data, averaged across pixels and color channels [9].

Due to their degenerate low-dimensional support, GANs do not define a density in the image space, which prevents measuring BPD on them. To endow a GAN with a full support and a likelihood, we train an inference network "around it", while keeping the weights of the GAN generator fixed. We also train an isotropic noise parameter $\sigma$. For both GANs and VAEs, we use this inference network to compute a lower bound to approximate the likelihood, *i.e.*, an upper bound on BPD. We evaluate all metrics using held-out data not used during training, which improves over common practice in the GAN literature, where training data is often used for evaluation.

## 5.1 Ablation study and comparison to VAE and GAN baselines

We conduct an ablation study on the CIFAR-10 dataset.[1] Our GAN baseline uses the non-residual architecture of SNGAN [37], which is stable and quick to train, without spectral normalization. The same convolutional architecture is kept to build a VAE baseline.[2] It produces the mean of a factorizing Gaussian distribution. To ensure a valid density model we add a trainable isotropic variance $\sigma$. We train the generator for coverage by optimizing $\mathcal{L}_Q(p_\theta)$, for quality by optimizing $\mathcal{L}_C(p_\theta)$, and for both by optimizing the sum $\mathcal{L}_Q(p_\theta) + \mathcal{L}_C(p_\theta)$. The model using Variational inference with Adaptive Density Estimation (ADE) is refered to as V-ADE. The addition of adversarial training is denoted AV-ADE, and hybrid training with a Gaussian decoder as AV-GDE. The bijective function $f_\psi$, implemented as a small Real-NVP with 1 scale, 3 residual blocks, 2 layers per block, increases the number of weights by approximately $1.4\%$. We compensate for these additional parameters with a slight decrease in the width of the generator for fair comparison.[3] See Appendix B for details.

Experimental results in Table 1 confirm that the GAN baseline yields better sample quality (IS and FID) than the VAE baseline: obtaining inception scores of $6.8$ and $2.0$, respectively. Conversely, VAE achieves better coverage, with a BPD of $4.4$, compared to $7.0$ for GAN. An identical generator trained for both quality and coverage, AV-GDE, obtains a sample quality that is in between that of the GAN and the VAE baselines, in line with the analysis in Section 4. Samples from the different models in Figure 5 confirm these quantitative results. Using $f_\psi$ and training with $\mathcal{L}_C(p_\theta)$ only, denoted by V-ADE in the table, leads to improved sample quality with IS up from $2.0$ to $3.0$ and FID down from $171$ to $112$. Note that this quality is still below the GAN baseline and our AV-GDE model.

When $f_\psi$ is used with coverage and quality driven training, AV-ADE, we obtain improved IS and FID scores over the GAN baseline, with IS up from $6.8$ to $7.1$, and FID down from $31.4$ to $28.0$. The

examples shown in the figure confirm the high quality of the samples generated by our AV-ADE model. Our model also achieves a better BPD than the VAE baseline. These experiments demonstrate that our proposed bijective feature space substantially improves the compatibility of coverage and quality driven training. We obtain improvements over both VAE and GAN in terms of held-out likelihood, and improve VAE sample quality to, or beyond, that of GAN. We further evaluate our model using the recent precision and recall approach of [43] an the classification framework of [48] in Appendix E. Additional results showing the impact of the number of layers and scales in the bijective similarity mapping $f_\psi$ (Appendix F), reconstructions qualitatively demonstrating the inference abilities of our AV-ADE model (Appendix G) are presented in the supplementary material.

## 5.2 Refinements and comparison to the state of the art

We now consider further refinements to our model, inspired by recent generative modeling literature. Four refinements are used: (i) adding residual connections to the discriminator [17] (rd), (ii) leveraging more accurate posterior approximations using inverse auto-regressive flow [26] (iaf); see Appendix B, (iii) training wider generators with twice as many channels (wg), and (iv) using a hierarchy of two scales to build $f_\psi$ (s2); see Appendix F. Table 2 shows consistent improvements with these additions, in terms of BPD, IS, FID.

| Refinements | BPD ↓ | IS ↑ | FID ↓ |
|---|---|---|---|
| GAN | [7.0] | 6.8 | 31.4 |
| GAN (rd) | [6.9] | 7.4 | 24.0 |
| AV-ADE | 3.9 | 7.1 | 28.0 |
| AV-ADE (rd) | 3.8 | 7.5 | 26.0 |
| AV-ADE (wg, rd) | 3.8 | **8.2** | **17.2** |
| AV-ADE (iaf, rd) | 3.7 | 8.1 | 18.6 |
| AV-ADE (s2) | **3.5** | 6.9 | 28.9 |

Table 2: Model refinements.

Table 3 compares our model to existing hybrid approaches and state-of-the-art generative models on CIFAR-10. In the category of hybrid models that define a valid likelihood over the data space, denoted by Hybrid (L) in the table, FlowGAN(H) optimizes MLE and an adversarial loss, and FlowGAN(A) is trained adversarially. The AV-ADE model significantly outperforms these two variants both in terms of BPD, from $4.2$ to between $3.5$ and $3.8$, and quality, *e.g.*, IS improves from $5.8$ to $8.2$. Compared to models that train an inference network adversarially, denoted by Hybrid (A), our model shows a substantial improvement in IS from $7.0$ to $8.2$. Note that these models do not allow likelihood evaluation, thus BPD values are not defined.

Compared to adversarial models, which are not optimized for support coverage, AV-ADE obtains better FID ($17.2$ down from $21.7$) and similar IS ($8.2$ for both) compared to SNGAN with residual connections and hinge-loss, despite training on $17\%$ less data than GANs (test split removed). The improvement in FID is likely due to this measure being more sensitive to support coverage than IS. Compared to models optimized with MLE only, we obtain a BPD between $3.5$ and $3.7$, comparable to $3.5$ for Real-NVP demonstrating a good coverage of the support of held-out data. We computed IS and FID scores for MLE based models using publicly released code, with provided parameters (denoted by $\dagger$ in the table) or trained ourselves (denoted by $\ddagger$). Despite being smaller (for reference Glow has 384 layers *vs*. at most 10 for our deeper generator), our AV-ADE model generates better samples, *e.g.*, IS up from $5.5$ to $8.2$ (samples displayed in Figure 6), owing to quality driven training controling where the off-dataset mass goes. Additional samples from our AV-ADE model and comparison to others models are given in Appendix A.

| Hybrid (L) | BPD ↓ | IS ↑ | FID ↓ |
|---|---|---|---|
| AV-ADE (wg, rd) | 3.8 | **8.2** | **17.2** |
| AV-ADE (iaf, rd) | 3.7 | 8.1 | 18.6 |
| AV-ADE (S2) | **3.5** | 6.9 | 28.9 |
| FlowGan(A) [16] | 8.5 | 5.8 | |
| FlowGan(H) [16] | 4.2 | 3.9 | |

| Hybrid (A) | BPD ↓ | IS ↑ | FID ↓ |
|---|---|---|---|
| AGE [52] | | 5.9 | |
| ALI [12] | | 5.3 | |
| SVAE [6] | | 6.8 | |
| $\alpha$-GAN [41] | | 6.8 | |
| SVAE-r [6] | | **7.0** | |

| Adversarial | BPD ↓ | IS ↑ | FID ↓ |
|---|---|---|---|
| mmd-GAN[1] | | 7.3 | 25.0 |
| SNGan [37] | | 7.4 | 29.3 |
| BatchGAN [32] | | 7.5 | 23.7 |
| WGAN-GP [17] | | 7.9 | |
| SNGAN$_{(R,H)}$ | | **8.2** | **21.7** |

| MLE | BPD ↓ | IS ↑ | FID ↓ |
|---|---|---|---|
| Real-NVP [9] | 3.5 | $4.5^\dagger$ | $56.8^\dagger$ |
| VAE-IAF [26] | 3.1 | $3.8^\dagger$ | $73.5^\dagger$ |
| Pixcnn++ [46] | **2.9** | 5.5 | |
| Flow++ [20] | 3.1 | | |
| Glow [24] | 3.4 | $5.5^\ddagger$ | $46.8^\ddagger$ |

Table 3: Performance on CIFAR10, without labels. MLE and Hybrid (L) models discard the test split. $\dagger$: computed by us using provided weights. $\ddagger$: computed by us using provided code to (re)train models.

## 5.3 Results on additional datasets

To further validate our model we evaluate it on STL10 ($48 \times 48$), ImageNet and LSUN (both $64 \times 64$). We use a wide generator to account for the higher resolution, without IAF, a single scale in $f_\psi$, and no residual blocks (see Section 5.2). The architecture and training hyper-parameters are not tuned, besides adding one layer at resolution $64 \times 64$, which demonstrates the stability of our approach. On STL10, Table 4 shows that our AV-ADE improves inception score over SNGAN, from $9.1$ up to

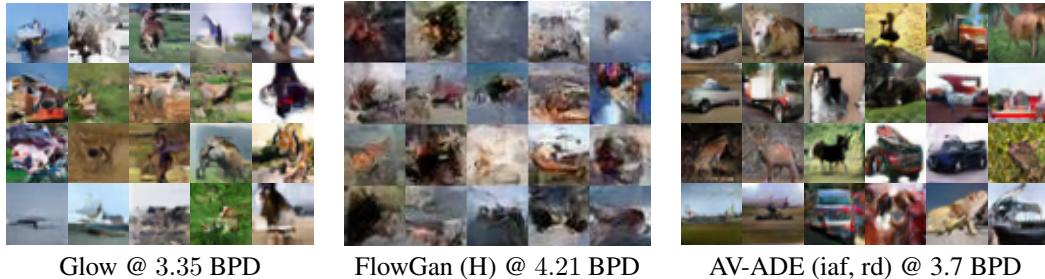

| Glow @ 3.35 BPD | FlowGan (H) @ 4.21 BPD | AV-ADE (iaf, rd) @ 3.7 BPD |

Figure 6: Samples from models trained on CIFAR-10. Our AV-ADE spills less mass on unrealistic samples, owing to adversarial training which controls where off-dataset mass goes.

9.4, and is second best in FID. Our likelihood performance, between 3.8 and 4.4, and close to that of Real-NVP at 3.7, demonstrates good coverage of the support of held-out data. On the ImageNet dataset, maintaining high sample quality, while covering the full support is challenging, due to its very diverse support. Our AV-ADE model obtains a sample quality behind that of MMD-GAN with IS/FID scores at 8.5/45.5 *vs.* 10.9/36.6. However, MMD-GAN is trained purely adversarially and does not provide a valid density across the data space, unlike our approach.

Figure 7 shows samples from our generator trained on a single GPU with 11 Gb of memory on LSUN classes. Our model yields more compelling samples compared to those of Glow, despite having less layers (7 *vs.* over 500). Additional samples on other LSUN categories are presented in Appendix A.

| STL-10 (48 × 48) | BPD ↓ | IS ↑ | FID ↓ | ImageNet (64 × 64) | BPD ↓ | IS ↑ | FID ↓ | LSUN | Real-NVP | Glow | Ours |
|---|---|---|---|---|---|---|---|---|---|---|---|
| AV-ADE (wg, wd) | 4.4 | **9.4** | 44.3 | AV-ADE (wg, wd) | 4.90 | 8.5 | 45.5 | Bedroom | (2.72/×) | (2.38/208.8[†]) | (3.91, **21.1**) |
| AV-ADE (iaf, wd) | 4.0 | 8.6 | 52.7 | Real-NVP | 3.98 | | | Tower | (2.81/×) | (2.46/214.5[†]) | (3.95, **15.5**) |
| AV-ADE (s2) | 3.8 | 8.6 | 52.1 | Glow | 3.81 | | | Church | (3.08/×) | (2.67/222.3[†]) | (4.3, **13.1**) |
| Real-NVP | **3.7**[‡] | 4.8[‡] | 103.2[‡] | Flow++ | **3.69** | | | Classroom | × | × | (4.6, 20.0) |
| BatchGAN | | 8.7 | 51 | MMD-GAN | | 10.9 | 36.6 | Restaurant | × | × | (4.7, 20.5) |
| SNGAN (Res-Hinge) | | 9.1 | **40.1** | | | | | | | | |

Table 4: Results on the STL-10, ImageNet, and LSUN datasets. AV-ADE (wg, rd) is used for LSUN.

| Glow [24] | Ours: AV-ADE (wg, rd) |

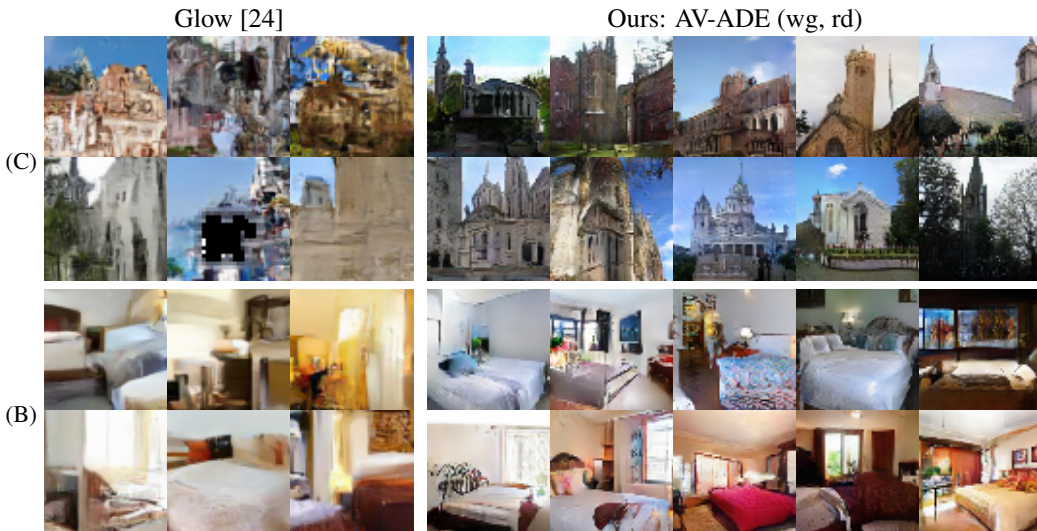

Figure 7: Samples from models trained on LSUN Churches (C) and bedrooms (B). Our AV-ADE model over-generalises less and produces more compelling samples. See Appendix A for more classes and samples.

# 6 Conclusion

We presented a generative model that leverages invertible network layers to relax the conditional independence assumptions commonly made in VAE decoders. It allows for efficient feed-forward sampling, and can be trained using a maximum likelihood criterion that ensures coverage of the data generating distribution, as well as an adversarial criterion that ensures high sample quality.

**Acknowledgments.** The authors would like to thank Corentin Tallec, Mathilde Caron, Adria Ruiz and Nikita Dvornik for useful feedback and discussions. Acknowledgments also go to our anonymous reviewers, who contributed valuable comments and remarks.

This work was supported in part by the grants ANR16-CE23-0006 "Deep in France", LabEx PERSYVAL-Lab (ANR-11-LABX0025-01) as well as the Indo-French project EVEREST (no. 5302-1) funded by CEFIPRA and a grant from ANR (AVENUE project ANR-18-CE23-0011),

## Footnotes

[1] We use the standard split of 50k/10k train/test images of 32×32 pixels. [2] In the VAE model, some intermediate feature maps are treated as conditional latent variables, allowing for hierarchical top-down sampling (see Appendix B). Experimentally, we find that similar top-down sampling is not effective for the GAN model.

[3] This is, however, too small to have a significant impact on the experimental results.

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
