[Supplementary Material · app.pdf]

# A Quantitative and qualitative results

In this section we provide additional quantitative and qualitative results on CIFAR10, STL10, LSUN (categories: Bedrooms, Towers, Bridges, Kitchen, Church, Living room, Dining room, Classroom, Conference room and Restaurant) at resolutions $64 \times 64$ and $128 \times 128$, ImageNet and CelebA. We report IS/FID scores together with BPD.

## A.1 Additional samples on CIFAR10

Samples from AV-ADE (wg, rd)          Real images

Figure 8: Samples from our AV-ADE (wg, rd) model trained on CIFAR10 compared to real images. A significant proportion of samples can be reasonably attributed to a CIFAR10 class, even though the model was trained without labels. Our model is constrained to cover the full support of the data, which translates to diverse samples, as noted in the qualitative results illustrated here.

## A.2 Quantitative results on LSUN categories

| LSUN ($64 \times 64$) | BPD↓ | FID↓ | LSUN ($128 \times 128$) | BPD↓ | FID↓ |
|---|---|---|---|---|---|
| Bedrooms | 3.92 | 21.1 | Bedrooms | 3.0 | 70.1 |
| Towers | 3.95 | 15.5 | Towers | 3.1 | 40.8 |
| Bridges | 4.1 | 25.2 | Bridges | 3.4 | 64.2 |
| Kitchen | 4.0 | 15.3 | Kitchen | 3.1 | 45.2 |
| Church | 4.3 | 13.2 | Church | 3.5 | 47.3 |
| Living | 4.1 | 16.7 | Living | 3.2 | 50.6 |
| Dining | 4.3 | 14.6 | Dining | 3.2 | 41.0 |
| Classroom | 4.6 | 20.0 | Classroom | 3.5 | 59.2 |
| Conf. room | 4.2 | 21.3 | Conf. room | 3.1 | 61.7 |
| Restaurant | 4.7 | 20.5 | Restaurant | 3.8 | 60.0 |

Table 5: Quantitative results obtained on LSUN categories, at resolutions $64 \times 64$ and $128 \times 128$ using our AV-ADE (wg, rd) model.

## A.3 Samples on all Lsun datasets

Figure 9: Samples obtained from our AV-ADE (wg, rd) model trained on LSUN bedrooms (left), compared to training images (right).

Figure 10: Samples obtained from our AV-ADE (wg, rd) model trained on LSUN towers (left), compared to training images (right).

Figure 11: Samples obtained from our AV-ADE (wg, rd) model trained on LSUN bridges (left), compared to training images (right).

Figure 12: Samples obtained from our AV-ADE (wg, rd) model trained on LSUN kitchen (left), compared to training images (right).

Figure 13: Samples obtained from our AV-ADE (wg, rd) model trained on LSUN churches (left), compared to training images (right).

Figure 14: Samples obtained from our AV-ADE (wg, rd) model trained on LSUN living rooms (left), compared to training images (right).

Figure 15: Samples obtained from our AV-ADE (wg, rd) model trained on LSUN dining rooms (left), compared to training images (right).

Figure 16: Samples obtained from our AV-ADE (wg, rd) model trained on LSUN conference rooms (left), compared to training images (right).

Figure 17: Samples obtained from our AV-ADE (wg, rd) model trained on LSUN restaurants (left), compared to training images (right).

## A.4 Additional samples on STL10

Samples from AV-ADE (wg, rd)                    Real images

Figure 18: Samples from our AV-ADE (wg, rd) model trained on STL10 compared to real images. Training was done without any label conditioning.

# B Model refinements

## B.1 Top-down sampling of hierarchical latent variables

Flexible priors and posteriors for the variational autoencoder model can be obtained by sampling hierarchical latent variables at different layers in the network. In the generative model $p_\theta$, latent variables $\mathbf{z}$ can be split into $L$ groups, each one at a different layer, and the density over $\mathbf{z}$ can be written as:

$$q(\mathbf{z}) = q(\mathbf{z}_L) \prod_{i=1}^{L-1} q(\mathbf{z}_i|\mathbf{z}_{i+1}). \tag{6}$$

Additionally, to allow the chain of latent variables to be sampled in the same order when encoding-decoding and when sampling, top-down sampling is used, as proposed in Bachman [3], Kingma et al. [26], Sønderby et al. [49]. With top-down sampling, the encoder (symmetric to the decoder) extracts deterministic features $h_i$ at different levels as the image is being encoded, constituting the bottom-up deterministic pass. While decoding the image, these previously extracted deterministic features $h_i$ are used for top-down sampling and help determining the posterior over latent variables at different depths in the decoder. These posteriors are also conditioned on the latent variables sampled at lower feature resolutions, using normal densities as follows:

$$
\begin{aligned}
q_\phi(\mathbf{z}_1|x) &= \mathcal{N}(\mathbf{z}_1|\mu_1(x, h_1), \sigma_1^2(x, h_1)), \\
q_\phi(\mathbf{z}_i|\mathbf{z}_{i-1}) &= \mathcal{N}(\mathbf{z}_i|\mu_i(x, \mathbf{z}_{i-1}, h_{i-1}), \sigma_i^2(x, \mathbf{z}_{i-1}, h_{i-1})).
\end{aligned}
$$

$$(7)$$
$$(8)$$

This constitutes the stochastic top-down pass. We refer the reader to Bachman [3], Kingma et al. [26], Sønderby et al. [49] for more detail.

## B.2 Inverse autoregressive flow

To increase the flexibility of posteriors used over latent variables in variational inference, Kingma et al. [26] proposed a type of normalizing flow called inverse autoregressive flow (IAF). The main benefits of this normalizing flow are its scalability to high dimensions, and its ability to leverage autoregressive neural network, such as those introduced by van den Oord et al. [53]. First, a latent variable vector is sampled using the reparametrization trick [25]:

$$\epsilon \sim \mathcal{N}(0, I), z_0 = \mu_0 + \sigma_0 \epsilon. \tag{9}$$

Then, mean and variance parameters $\mu_1$ and $\sigma_1$ are computed as functions of $z_0$ using autoregressive models, and a new latent variable $z_1$ is obtained:

$$z_1 = \mu_1(z_0) + \sigma_1(z_0)z_0. \tag{10}$$

Since $\sigma_1$ and $\mu_1$ are implemented by autoregressive networks, the Jacobian $\frac{dz_1}{dz_0}$ is triangular with the values of $\sigma_1$ on the diagonal, and the density under the new latent variable remains efficient to compute. This transformation can be repeated an arbitrary number of times for increased flexibility in theory, but in practice a single step is used.

## B.3 Gradient penalty

A body of work on generative adversarial networks centers around the idea of regularizing the discriminator by enforcing Lipschitz continuity, for instance by Arjovsky et al. [2], Gulrajani et al. [17], Miyato et al. [37], Thanh-Tung et al. [51]. In this work, we use the approach of Gulrajani et al. [17], that enforces the Lipschitz constraint with a gradient penalty term added to the loss:

$$\mathcal{L}_{Grad} = \lambda + \mathbb{E}_{\hat{x}}[(||\Delta_{\hat{x}} D(\hat{x})||_2 - 1)^2], \tag{11}$$

where $\hat{x}$ is obtained by interpolating between real and generated data:

$$\epsilon \quad \sim \quad U_{[0,1]}, \tag{12}$$
$$\hat{x} \quad = \quad \epsilon x + (1 - \epsilon)\tilde{x}. \tag{13}$$

We add this term to the loss used to train the discriminator that yields our quality driven criterion.

## C Implementation details

**Architecture and training hyper-parameters.** We used Adamax [23] with learning rate 0.002, $\beta_1 = 0.9$, $\beta_2 = 0.999$ for all experiments. The convolutional architecture of the models used is described in Table 6a and Table 6b. All CIFAR-10 experiments use batch size 64, other experiments in high resolution use batch size 32. To stabilize the adversarial training we use the gradient penalty [17] with coefficient 100, and 1 discriminator update per generator update. We experimented with different weighting coefficients between the two loss components, and found that values in the range 10 to 100 on the adversarial component work best in practice. No significant influence on the final performance of the model is observed in this range, though the training dynamics in early training are improved with higher values. With values significantly smaller than 10, discriminator collapse was observed in a few isolated cases. All experiments reported here use coefficient 100.

For experiments with hierarchical latent variables, we use 32 of them per layer. In the generator we use ELU nonlinearity, in discriminator with residual blocks we use ReLU, while in simple convolutional discriminator we use leaky ReLU with slope 0.2.

Unless stated otherwise we use three Real-NVP layers with a single scale and two residual blocks that we train only with the likelihood loss. Regardless of the number of scales, the VAE decoder always outputs a tensor of the same dimension as the target image, which is then fed to the Real-NVP layers. As in the reference implementations, we use both batch normalization and weight normalization in Real-NVP and only weight normalization in IAF. We use the reference implementations of IAF and Real-NVP released by the authors.

| Discriminator | Generator |
|---|---|
|  | conv $3 \times 3$, 16 |
|  | IAF block 32 |
| conv $3 \times 3$, 16 | IAF block down 64 |
| ResBlock 32 | IAF block down 128 |
| ResBlock down 64 | IAF block down 256 |
| ResBlock down 128 | $h \sim \mathcal{N}(0;1)$ |
| ResBlock down 256 | IAF block up 256 |
| Average pooling | IAF block up 128 |
| dense 1 | IAF block up 64 |
|  | IAF block 32 |
|  | conv $3 \times 3$, 3 |

Table 6: Residual architectures for experiments from Section 5.2 and Table 9

## D   On the Inception Score and the Fréchet inception distance

Quantitative evaluation of Generative Adversarial Networks is challenging, in part due to the absence of log-likelihood. Inception score (IS) and Fréchet Inception distance (FID) are two measures proposed by [45] and [19] respectively, to automate the qualitative evaluation of samples. Though imperfect, they have been shown to correlate well with human judgement in practice, and it is standard in GAN literature to use them for quantitative evaluations. These metrics are also sensitive to coverage. Indeed, any metric evaluating quality only would be degenerate, as collapsing to the mode of the distribution would maximize it. However, in practice both metrics are much more sensitive to quality than to support coverage, as we evaluate below.

**Inception score (IS)** [45] is a statistic of the generated images, based on an external deep network trained for classification on ImageNet. It is given by:

$$IS(p_\theta) = \exp\left(\mathbb{E}_{x \sim p_\theta} \mathcal{D}_{\mathrm{KL}}\left(p\left(y|x\right) || p(y)\right)\right), \tag{14}$$

where $x \sim p_\theta$ is sampled from the generative model, $p(y|x)$ is the conditional class distribution obtained by applying the pretrained classification network to the generated images, and $p(y) = \int_x p(y|x) p_\theta(x)$ is the class marginal over the generated images.

**Fréchet Inception distance (FID)** [19] compares the distributions of Inception embeddings *i.e.*, activations from the penultimate layer of the Inception network, of real ($p_r(\mathbf{x})$) and generated ($p_g(\mathbf{x})$) images. Both these distributions are modeled as multi-dimensional Gaussians parameterized by their respective mean and covariance. The distance measure is defined between the two Gaussian distributions as:

$$d^2((\mathbf{m}_r, \mathbf{C}_r), (\mathbf{m}_g, \mathbf{C}_g)) = \|\mathbf{m}_r - \mathbf{m}_g\|^2 + \mathrm{Tr}(\mathbf{C}_r + \mathbf{C}_g - 2(\mathbf{C}_r \mathbf{C}_g)^{\frac{1}{2}}), \tag{15}$$

where $(\mathbf{m}_r, \mathbf{C}_r)$, $(\mathbf{m}_g, \mathbf{C}_g)$ denote the mean and covariance of the real and generated image distributions respectively.

**Practical use.** IS and FID correlate predominantly with the quality of samples. In GAN literature, for instance Miyato et al. [37], they are considered to correlate well with human judgement of quality. An empirical indicator of that is that state-of-the art likelihood-based models have very low IS/FID scores despite having good coverage, which shows that the low quality of their samples impacts IS/FID more heavily than their coverage performance. Conversely, state-of-the art adversarial models have high IS/FID scores, despite suffering from mode dropping (which strongly degrades BPD). So the score is determined mainly by the high quality of their samples. This is especially true when identical architectures and training budget are considered, as in our first experiment in Section 5.1.

To obtain a quantitative estimation of how much entropy/coverage impacts the IS and FID measures, we evaluate the scores obtained by random subsamples of the dataset, such that the quality is unchanged but coverage progressively degrades (see details of the scores below). Table 7 shows that when using the full set of images (50k) the FID is 0 as the distributions are identical. Notice that as the number of images decreases, IS is very stable (it can even increase, but by very low increments that fall below statistical noise, with a typical standard deviation of 0.1). This is because the entropy of the

| Split size | IS | FID |
|---|---|---|
| 50k (full) | 11.3411 | 0.00 |
| 40k | 11.3388 | 0.13 |
| 30k | 11.3515 | 0.35 |
| 20k | 11.3458 | 0.79 |
| 10k | 11.3219 | 2.10 |
| 5k | 11.2108 | 4.82 |
| 2.5k | 11.0446 | 10.48 |

Table 7: IS and FID scores obtained by the ground truth when progressively dropping parts of the dataset. The metrics are largely insensitive to removing most of the dataset, unlike BPD. For reference, a reasonable GAN could get around 8 IS and 20 FID.

distribution is not strongly impacted by subsampling, even though coverage is. FID is more sensitive, as it behaves more like a measure of coverage (it compares the two distributions). Nonetheless, the variations remain extremely low even when dropping most of the dataset. For instance, when removing $80\%$ of the dataset (*i.e.*, using 10k images), FID is at 2.10, to be compared with typical GAN/AV-ADE values that are around 20. These measurements demonstrate that IS and FID scores are heavily dominated by the quality of images. From this, we conclude that IS and FID can be used as reasonable proxies to asses sample quality, even though they are also slightly influenced by coverage. One should bear in mind, however, that a small increase in these scores may come from better coverage rather than improved sample quality.

# E  Other evaluation metrics

## E.1  Evaluation using samples as training data for a discriminator

We also evaluate our approach using the two metrics recently proposed by Shmelkov et al. [48]. This requires a class-conditional version of AV-ADE. To address the poor compatibility of conditional batch-normalization with VAEs, we propose conditional weight normalization (CWN), see below for details. Apart from this adaptation, the architecture is the same as in Section 5.1.

The first metric, GAN-test, is obtained by training a classifier on natural image data and evaluating it on generated samples. It is sensitive to sample quality only. Our AV-ADE model obtains a slightly higher GAN-test score, suggesting comparable sample quality, which is in line with the results in Section 5.1.

| model | GAN-test (%) | GAN-train (%) |
|---|---|---|
| GAN | 71.8 | 29.7 |
| AV-ADE | **76.9** | **73.4** |
| DCGAN[†] | 58.2 | 65.0 |

Table 8: GAN-test and GAN-train measures for class conditional CQFG and GAN models on CIFAR-10. The performance of the DCGAN[†] model, though not directly comparable, is provided as a reference point.

The second metric, GAN-train, is obtained by training a classifier on generated samples and evaluating it on natural images. Having established similar GAN-test performance, this demonstrates improved sample diversity of the AV-ADE model and shows that the coverage-driven training improves the support of the learned model.

**Class conditioning with conditional weight-normalization.**  To perform this evaluation we develop a class conditional version of our AV-ADE model. The discriminator is conditioned using the class conditioning introduced by Miyato and Koyama [36]. GAN generators are typically made class-conditional using conditional batch normalization [8, 13], however batch normalization is

known to be detrimental in VAEs [26], as we verified in practice. To address this issue, we propose conditional weight normalization (CWN). As in weight normalization [44], we separate the training of the scale and the direction of the weight matrix. Additionally, the scaling factor $g(y)$ of the weight matrix $\mathbf{v}$ is conditioned on the class label $y$:

$$\mathbf{w} = \frac{g(y)}{\|\mathbf{v}\|}\mathbf{v}, \tag{16}$$

We also make the network biases conditional on the class label. Otherwise, the architecture is the same one used for the experiments in Section 5.1.

### E.2 Model evaluation using precision and recall

In this section, we evaluate our models using the precision and recall procedure of Sajjadi et al. [43]. This evaluation is relevant as it seeks to evaluate coverage of the support and the quality of the samples separately, rather than aggregating them into a single score. Intuitively, the precision measures the sample quality of the model, while the recall measures to what extent the model covers the support of the target distribution. The precision-recall metrics are determined based on a clustering of *Pool3* features extracted from training images and generated images using a pre-trained Inception network. The resulting histograms over cluster assignments are then used to assess the similarity of the distributions.

Figure 19: Precision-recall curves using the evaluation procedure of [43].

Figure 19 presents the evaluation of our models in Section 5.1, as well as the Glow and Real-NVP models, using the official code provided online by the authors at `https://github.com/msmsajjadi/precision-recall-distributions`. Our AV-ADE model obtains a better area under curve (AUC) than the GAN baseline, and model refinements improve AUC further. For comparison, Glow and Real-NVP have lower AUC than both GAN and our models.

## F   Qualitative influence of the feature space flexibility

We experiment with different architectures to implement the invertible mapping used to build the feature space as presented in Section 4. To assess the impact of the expressiveness of the invertible model on the behavior of our framework, we modify various standard parameters of the architecture. Popular invertible models such as Real-NVP [9] readily offer the possibility of extracting latent representation at several scales, separating global factors of variations from low level detail. Thus, we experiment with varying number of scales. Another way of increasing the flexibility of the model is to change the number of residual blocks used in each invertible layer. Note that all the models evaluated so far in the main body of the paper are based on a single scale and two residual blocks, except the one denoted with (s2). In addition to our AV-ADE models, we also compare with similar models trained with maximum likelihood estimation (V-ADE). Models are first trained with maximum-likelihood estimation, then with both coverage and quality driven criteria.

The results in Table 9 show that factoring out features at two scales rather than one is helpful in terms of BPD. For the AV-ADE models, however, IS and FID deteriorate with more scales, and so a tradeoff between must be struck. For the V-ADE models, the visual quality of samples also improves when using multiple scales, as reflected in better IS and FID scores. Their quality, however, remains far worse than those produced with the coverage and quality training used for the AV-ADE models. Samples in the maximum-likelihood setting are provided in Figure 20. With three or more scales, models exhibit symptoms of overfitting: train BPD keeps decreasing while test BPD starts increasing, and IS and FID also degrade.

| Scales | Blocks | BPD ↓ | IS ↑ | FID ↓ |
|--------|--------|-------|------|-------|
| 1 | 2 | 3.77 | **7.9** | **20.1** |
| 2 | 2 | 3.48 | 6.9 | 27.7 |
| 2 | 4 | **3.46** | 6.9 | 28.9 |
| 3 | 3 | 3.49 | 6.5 | 31.7 |

(a) AV-ADE models

| Scales | Blocks | BPD ↓ | IS ↑ | FID ↓ |
|--------|--------|-------|------|-------|
| 1 | 2 | 3.52 | 3.0 | 112.0 |
| 2 | 2 | **3.41** | **4.5** | 85.5 |
| 3 | 2 | 3.45 | 4.4 | **78.7** |
| 4 | 1 | 3.49 | 4.1 | 82.4 |

(b) V-ADE models

Table 9: Evaluation on CIFAR-10 of different architectures of the invertible layers of the model.

No $f_\psi$

$f_\psi$ 1 scale

$f_\psi$ 2 scales

$f_\psi$ 3 scales

Figure 20: Samples obtained using VAE models trained with MLE (Table 9b) showing qualitative influence of multi-scale feature space. The models include one without invertible decoder layers, and with real-NVP layers using one, two and three scales. The samples illustrate the impact of using invertible real-NVP layers in these autoencoders.

# G   Visualisations of reconstructions

We display reconstructions obtained by encoding and then decoding ground truth images with our models (AV-ADE from Section 5.1) in Figure 21. As is typical for expressive variational autoencoders, real images and their reconstructions cannot be distinguished visually.

Real image

AV-ADE reconstruction

Figure 21: Real images and their reconstructions with the AV-ADE models.

# H   Coverage and Quality driven training algorithm

---

**Algorithm 1** Coverage and Quality driven training for our AV-ADE model.

---

**for** number of training steps **do**

- Sample $m$ real images $\{\boldsymbol{x}^{(1)}, \ldots, \boldsymbol{x}^{(m)}\}$ from $p^*$, approximated by the dataset.
- Map the real images to feature space $\{f(\boldsymbol{x}^{(1)}), \ldots, f(\boldsymbol{x}^{(m)})\}$ using the invertible transformation $f$.
- Encode the feature space vectors using the VAE encoder and get parameters for the posterior $q_\phi(\boldsymbol{z}|f(\boldsymbol{x}))$.
- Sample $m$ latent variable vectors, $\{\hat{\boldsymbol{z}}^{(1)}, \ldots, \hat{\boldsymbol{z}}^{(m)}\}$ from the posterior $q_\phi(z|x)$, and $m$ latent variable vectors $\{\tilde{\boldsymbol{z}}^{(1)}, \ldots, \tilde{\boldsymbol{z}}^{(m)}\}$ from the VAE prior $p_\theta(z)$
- Decode both sets of latent variable vectors using the VAE decoder into the means of conditional Gaussian distributions, $\{\mu(\hat{\boldsymbol{z}})^{(1)}, \ldots, \mu(\hat{\boldsymbol{z}})^{(m)}\}$ and $\{\mu(\tilde{\boldsymbol{z}})^{(1)}, \ldots, \mu(\tilde{\boldsymbol{z}})^{(m)}\}$
- Sample from the Gaussian densities obtained, $\{\mathcal{N}(.|\mu(\hat{\boldsymbol{z}})^{(i)}, \sigma I_n)\}_{i \le m}$ and $\{\mathcal{N}(.|\mu(\tilde{\boldsymbol{z}})^{(i)}, \sigma I_n)\}_{i \le m}$, which yields reconstructions in feature space $\{\widehat{f(\boldsymbol{x})}^{(i)}\}_{i \le m}$ and samples in feature space $\{\widetilde{f(\boldsymbol{x})}^{(i)}\}_{i \le m}$
- Map the samples and reconstructions back to image space using the inverse of the invertible transformation $f^{-1}$ which yields reconstructions $\{\hat{\boldsymbol{x}}^{(i)}\}_{i \le m}$ and samples $\{\tilde{\boldsymbol{x}}^{(i)}\}_{i \le m}$
- Compute $\mathcal{L}_C(p_\theta)$ using ground truth images $\{\boldsymbol{x}^{(i)}\}_{i \le m}$ and their reconstructions $\{\hat{\boldsymbol{x}}^{(i)}\}_{i \le m}$
- Compute $\mathcal{L}_Q(p_\theta)$ by feeding the ground truth images $\{\boldsymbol{x}^{(i)}\}_{i \le m}$ together with the sampled images $\{\tilde{\boldsymbol{x}}^{(i)}\}_{i \le m}$ to the discriminator
- Optimize the discriminator by gradient descent to bring $\mathcal{L}_Q$ closer to $\mathcal{L}_Q^*$
- Optimize the generator by gradient descent to minimize $\mathcal{L}_Q + \mathcal{L}_C$

**end for**

---