[Reviews · NeurIPS 2019]

Reviewer 1



I enjoyed reading this paper because it presents a number of important points that are problematic when it comes to deep generative models in a clear manner. In the related work section, I was missing work on DGM evaluated in feature space, e.g. [1]. Although the approach here has more mathematical rigor, it would be nice to acknowledge that the idea is not totally new. In the method section, I was not 100 % happy with the introduction of the feature space. Conceptually, what is happening is that you train a structure that looks like this, right: VAE - data layer - flow-based generative model - GAN. One could argue that the flow-based generative model alone already suffices to generate data. Is the advantage that the VAE does the generative work and the inverse-layers do not have to be as powerful? Please provide a more rigorous discussion around this and point to relevant related work. The experiments include enough datasets and the appendix shows actually images that are large enough to see some detail. Originality: high - both with respect to presentation of the facts and the method Quality: high - well written Clarity: good Significance: good [1] Hou, Xianxu, et al. "Deep feature consistent variational autoencoder." 2017 IEEE Winter Conference on Applications of Computer Vision (WACV). IEEE, 2017.

Reviewer 2



Summary: The authors propose a hybrid method that combines VAEs with adversarial training and flow based models. In particular, they derive an explicit density function p(x) where the likelihood can be evaluated, the corresponding components p(x|z) are more flexible than the standard VAE that utilizes diagonal Gaussians, and the generated samples have better quality than a standard VAE. The basic idea of the proposed model is that the VAE is defined between a latent space and an intermediate representation space, and then, the representation space is connected with the data space through an invertible non-linear flow. In general, I think the paper is quite well written, but on the same time I believe that there is a lot of compressed information, and the consequence is that in some parts it is not even clear what the authors want to say (see Clarity comments). The proposed idea of the paper seems quite interesting, but on the same time I have some doubts (see Quality comments). Clarity: 1. My main concern is that a lot of information is presented in a compressed way. For example, in Sec. 3 the general drawbacks of the VAEs and GANs are presented, and later in Sec. 4 a potential solution is provided. However, I think that is not crystal clear from the text how the proposed method solves these problems. For instance, one of the basic ideas is that the proposed method intuitively produces more flexible components p(x|z), such that to pull the density towards the image manifold. This is the main improvement upon the VAE, however, this is not mentioned clearly in the text. In general, I believe that the text is a bit "wordy", and instead, it should be more "to the point". 2. The term "strictly proper" is introduced but not used later in the text (line 102). 3. In my opinion the Fig. 1 & Fig. 2 are not so informative. 4. In Eq. 2 I think the p_\theta(x) is missed in the begging. Also, in Eq. 5 real data should not be used during the training of the discriminator? 5. In line 196 the optimal discriminator is refereed to Eq. 5, but this is not clear from the text. 6. In line 201, the two KL divergences are not symmetric, but the sum of them is. 7. The invertible function is not discussed in the main paper at all. Of course, information is provided in the appendix, but since the main paper should be "standalone" and the invertible function is a fundamental component of the method, I think that some discussion should be included. 8. In the Table 3 (MLE), how the BPD of NVP, VAE-IAF, Glow are computed, since for the IS and FID you use some symbols that refer to notes. 9. What the $\times$ means in Table 4? Also, what the LSUN table shows? 10. The NVP is the same with Real-NVP? Because both names are used. 11. I think that in the discussion of the experiments, the reasons and the intuition should be stated that explain why the proposed method is better than the competitive methods. In my opinion the current discussion does not explain the benefits of the proposed method. Quality: To my understanding, the basic idea of the paper is to define a VAE between a latent space and a representation space, and then connect the representation space with the data space through an invertible non-linear function f. Then, an adversarial training is applied to improve the sample quality. 1. I think that the goal of this approach is to construct more expressive/flexible p(x|z), which will pull the pdf mass closer to the data manifold. Instead, the diagonal Gaussian puts a lot of pdf mass outside of the data manifold due to the conditional independence assumption and the actual non-linear structure of the data manifold locally. So if an adversarial training is applied, the diagonal Gaussian constraint of the standard VAE will "regularize" to some extend the mu(z) and this will not improve the samples. While, using the proposed method with the flexible p(x|z), the mass can be pulled easier towards the manifold using the adversarial training. Is this explanation correct? If yes, then this makes sense, however from the text this is not communicated properly. 2. My main concern is that the method seems to be very similar to the Real-NVP. In particular, I do not see where the middle representation helps. Since, simply a Real-NVP can be used, which will produce a complex p(x) potentially closer to the image manifold, and then the adversarial training could be utilized to improve the quality of the samples. In particular, in Eq. 5 the f^-1(\mu(z)) can be seen essentially just as an NVP flow. So the question is: why the proposed method is better than a Real-NVP with an additional adversarial training? 3. In the Abstract is mentioned that the proposed method "is more efficient than fully invertible models". Why is this the case? I think that the representation space and the image space have the same dimensionality, so the invertible function will have the same size as other flow models. Based on my comment 2, why your method is more efficient than a Real-NVP with an additional adversarial training? Since the proposed model needs additionally to the invertible non-linear function f the mean and variance functions of the VAE encoder and decoder. So there are more parameters to be trained. Originality: The related work is cited properly. As the authors state, the combination of VAE with GANs is not a new idea. Of course, the proposed trick with the intermediate stochastic representation seems to be novel. However, as I commented in Quality, I am not sure how essential is the difference between the proposed approach and a Real-NVP with an additional adversarial training. Significance: The experimental results follow the proposed theory. In particular, in Sec. 5.1 is shown how each component of the method helps i.e. the flow improves the likelihood, the adversarial training the sampling quality and both of them together. Also, compared to other similar and state of the art models, the proposed model achieves higher log-likelihoods and improved sampling quality. However, I think that a good baseline experiment is the Real-NVP including the adversarial training, because in my opinion this is the closest model. Moreover, I would like to see some toy examples that present the p(x) differences between a VAE and the proposed model, as well as the differences between the p(f(x)) and the prior p(z)? In particular, I would like to see if the p(f(x)) learns indeed a useful representation, or if it is similar to a Gaussian. ---------------------------------------------------------------------------------------------------------------------------- After reading the rest of the reviews and the author’s response, I will improve my score to 5. However, I still believe that the paper can be improved further as regards its clarity, especially in Sec. 3 and Sec. 4 and Fig. 1, 2, 3.

Reviewer 3



The main idea of the proposed model is to introduce an intermediate feature space between the image space and the latent space. The image and feature spaces are linked by an intervertible neural network. The authors formulate the ELBO of a VAE between the latent and the feature space, and forward it to the image space using the change of variable formula. By considering the expected posterior in the feature space, the resulting architecture allows efficient sampling, which permits to add an adversarial loss in the image space. - Originality and significance: This paper is novel and impactful. Bridging the gap between GANs and density modelling networks and taking the best of both worlds is an important contribution. - Clarity: Writing is very good and clear. - Quality: The paper clearly demonstrates the importance of its proposition. Ablation experiments and comparison with other networks on multiple datasets are compelling. In particular, both the Inception Score and Fréchet distance are at least as good as the state-of-the-art on all tasks, up to a slightly larger bits per dimension score.

[Author Response · NeurIPS 2019]

We would like to thank the reviewers for their positive feedback, constructive comments and suggestions. Below we respond to the main points raised and will update the paper to take them into account.

**Response to Reviewer 1**

**R1.1: I was missing work on DGM evaluated in feature space, e.g. [1].** Thank you for pointing out this additional reference, we will include it in our discussion of related work. It is indeed related in the sense of using L2 reconstruction losses in abstract feature spaces, rather than in RGB pixel space. They rely on pre-trained ImageNet classification networks to define the feature space. In our work, the feature space is based on invertible transformations, which permits to train the feature space itself in a unified unsupervised learning framework, by optimizing the data likelihood.

**R1.2: Is the advantage that the VAE does the generative work and the inverse-layers do not have to be as powerful?** Yes, this is precisely the point. The "cheap", but non-invertible, VAE layers are more efficient in the sense that they update all the pixels in every layer, rather than half the pixels in each "coupling layer" of an NVP (each of which consists itself of many CNN layers). This makes the VAE layers mix pixel information more quickly. The VAE can be seen as a "fancy trained prior" for the NVP layers, rather than using a standard normal prior, thus requiring less coupling layers to obtain the desired density model. Equivalently, the NVP layers play the role of a "fancy noise model" that improves over the factored Gaussian over RGB values used in vanilla VAEs. Please see also response R2.3 below.

**Response to Reviewer 2**

**R2.1: Clarity.** Thank you for the useful and detailed comments on improving clarity. We will revise the text accordingly, taking these into account one by one. Here we briefly respond to a selection of them for sake of brevity. (1) We will make the link between Sections 3 and 4 more explicit and synthetic. (3) We will improve the captions of figures 1 and 2. (8) BPD is reported in the original papers and was matched exactly when retraining the models ourselves. We will also move the explanation of † and ‡ to the caption of Table 3. (10) We will ensure uniform usage of the terms NVP. (11) More interpretation of the ablation study (Section 5.1) and model refinements (Sec. 5.2 paragraph 1) will be provided.

**R2.2: Ablations.** – *"[...] So there are more parameters to be trained.":* We would like to first clarify that the invertible transformation $f_\psi$ used in our model is very light. In short, it uses three affine coupling layers, parametrised using two residual blocks rather than eight, implemented as in Real-NVP. In Table 1, the number of weights in the VAE is adjusted to account for the weights in $f_\psi$, resulting in models with the same number of weights across Table 1. See also line 231 (main paper) and Appendix B. As suggested in Clarity (7), we will make the main text self-contained regarding $f_\psi$.
– *"[...] A good baseline experiment is the Real-NVP including the adversarial training":* Definitely, in fact the Flow-GAN (ref. [17] in the main paper) trained with hybrid losses provides this baseline. We will emphasize this in Section 5.2 paragraph 2 (please see this section for details). We experimentally compare to Flow-GAN in Table 3 and Figure 6. Results show that we obtain a substantial improvement both in BPD and in sample quality. Note that in our first ablation study (Table 1), simply removing the VAE component results in a tiny flow model (see above), while growing it to be comparable yields the same as the Flow-Gan model.

**R2.3: Interpretation.** *"Why is your method more efficient[/better] than a Real-NVP with [...] adversarial training?"* The constraint of invertibility in the data space can be costly. Instead, VAEs are able to focus on the low dimensional manifold of natural images, at the cost of approximate inference. Invertible layers used by flow models in practice, affine transformations of half the variables, are quite restrictive (see also response R1.2 above). This is mitigated with expensive residual blocks to parametrize them. From that perspective, using mostly cost-efficient layers from the VAE and a few invertible layers where needed is a cost-efficient choice, which tends to be confirmed by experimental results.

**Response to Reviewer 3**

**R3.1: Would it be possible to improve the compression efficiency of the proposed model?** Good question. Without resorting to autoregressive models, which suffer from slow sampling, there are a number of ways to further improve the BPD of our models. (i) The most straightforward is to use "more muscle": training bigger models, and train for more iterations. In the presented experiments, all the models have been trained on a single consumer grade GPUs. (ii) It was recently shown in the Flow++ (ref. [22] in the main paper) that variational dequantization of the discrete pixel values can lead to significant improvements in BPD compared to uniform dequantization. This can also be applied to our model. (iii) Another possibility is to use a discrete flow component, such as [Integer Discrete Flows and Lossless Compression. arXiv:1905.0737] and use a mixture of logistic density functions to model the discrete target features.

**R3.2: Would it be feasible to sample from the true distribution for the GAN update?** Thanks for raising this question. Yes, this is definitely possible. We experimented with this before submitting the paper, but found the difference in results to be insignificant from using the mean. We will comment on this point in the final version of the paper, and add these results to the supplementary material.

[Meta-Review · NeurIPS 2019]

This paper proposes a new hybrid generative model, combining a maximum-likelihood approach with GANs. The authors are to be commended for their practical and conceptually interesting work. In the final version, the paper would also benefit from a discussion of [1], related work that introduces an alternative maximum likelihood perspective of GANs, and provides Bayesian generalizations. [1] Saatchi, Y and Wilson, A.G. Bayesian GAN. NeurIPS 2017. https://arxiv.org/abs/1705.09558